# Tegoprazan–Amoxicillin Dual Therapy for Clarithromycin-Resistant *Helicobacter pylori*: A Feasibility Pilot Study

**DOI:** 10.3390/microorganisms13102408

**Published:** 2025-10-21

**Authors:** Jun-Hyung Cho

**Affiliations:** Digestive Disease Center, Soonchunhyang University Hospital, 59, Daesagwan-ro, Yongsan-gu, Seoul 04401, Republic of Korea; chojhmd@naver.com

**Keywords:** Tegoprazan, amoxicillin, dual therapy, *Helicobacter pylori*, eradication

## Abstract

Tegoprazan (TPZ) has the potential to enhance *Helicobacter pylori* eradication. This study aimed to investigate the efficacy of TPZ–amoxicillin (TA) dual therapy against clarithromycin-resistant *H. pylori* strains. All *H. pylori*-positive patients were diagnosed by real-time polymerase chain reaction that can detect point mutations causing clarithromycin resistance. Patients infected with clarithromycin-resistant *H. pylori* received TA dual therapy consisting of 50 mg TPZ twice daily and high-dose amoxicillin (3 g per day) for 2 weeks. A total of 57 patients received TA dual therapy. There was a significant difference in *H. pylori* eradication rates between the treatment-naïve (*n* = 40) and treatment-experienced (*n* = 17) groups in the intention-to-treat and per-protocol analyses (60.0% vs. 29.4%, *p* = 0.035 and 70.6% vs. 31.3%, *p* = 0.009, respectively). Compliance with the eradication regimen was 100%, with an 8% incidence of adverse events. Multivariate analysis revealed that treatment-naïve status was a significantly independent factor associated with *H. pylori* eradication success (odds ratio, 8.679; *p* = 0.007). In conclusion, the eradication efficacy of TA dual therapy against clarithromycin-resistant *H. pylori* strain infections was suboptimal in treatment-naïve patients. Notably, eradication rates were significantly lower in patients with a prior history of *H. pylori* treatment.

## 1. Introduction

For decades, standard triple therapy consisting of a proton pump inhibitor (PPI), amoxicillin, and clarithromycin has been the mainstay of first-line *Helicobacter pylori* eradication. However, the effectiveness of this regimen has markedly declined, mainly due to an increase in the prevalence of antibiotic-resistant *H. pylori* strains [1]. Clarithromycin resistance is a major cause of *H. pylori* eradication failure in patients receiving the standard triple therapy [2]. International guidelines have recommended the avoidance of clarithromycin-containing treatment regimens in regions where clarithromycin resistance exceeds 15% owing to a strong association between clarithromycin resistance and eradication failure [3]. In the U.S., clarithromycin resistance rates range from 15.2% to 24.9% [4]. In most Asian countries, clarithromycin resistance rates in *H. pylori* have increased significantly over the past decade, with the pooled prevalence rate rising from approximately 29.1% during 2015–2019 to approximately 36.5% in 2020–2023 [5]. Therefore, bismuth quadruple or concomitant therapy has been recommended as a first-line treatment in regions with high clarithromycin resistance [6].

Amoxicillin is a broad-spectrum antibiotic that plays a critical role in the treatment of *H. pylori* infections. Unlike clarithromycin, metronidazole, and levofloxacin, amoxicillin resistance among *H. pylori* strains remains relatively low (5–10%) [7]. Recently, dual therapy combining frequent amoxicillin administration (e.g., 750 mg four times daily or 1000 mg three times daily) with potent acid suppression has emerged as an effective regimen for *H. pylori* eradication [8]. Dual therapy requires the administration of high doses of second-generation PPIs or vonoprazan to maintain a nearly neutral intragastric pH throughout the day, thereby enhancing the stability and bactericidal activity of amoxicillin against *H. pylori* [9]. Tegoprazan (TPZ), a novel potassium-competitive acid blocker (P-CAB), provides rapid, strong, and sustained gastric acid suppression, comparable to that of vonoprazan [10]. However, limited data are available on the treatment outcomes of TPZ–amoxicillin (TA) dual therapy in relation to clarithromycin resistance in *H. pylori*. Therefore, this pilot study was designed to evaluate the efficacy and safety of TA dual therapy for eradicating clarithromycin-resistant *H. pylori* strains.

## 2. Materials and Methods

### 2.1. Patients and Study Design

Patients were enrolled at a tertiary university hospital in Seoul, South Korea, between November 2024 and April 2025. All patients underwent gastroscopy for the evaluation of upper abdominal symptoms, endoscopic treatment of early-stage gastric neoplasia, or gastric cancer screening. Prior to endoscopy, the patients completed a questionnaire regarding their history of *H. pylori* eradication, family history of gastric cancer, weight, height, alcohol consumption, smoking, and the presence of comorbidities. All patients provided written informed consent for the endoscopic procedure and gastric biopsy for *H. pylori* testing. During endoscopy, the diagnosis of an *H. pylori* infection was confirmed by employing real-time polymerase chain reaction (RT-PCR). Based on RT-PCR results, patients infected with clarithromycin-resistant *H. pylori* strains were invited to participate in this study. The exclusion criteria were a history of gastric surgery, severe systemic disease, refusal to undergo *H. pylori* eradication treatment, or other known causes.

### 2.2. Helicobacter pylori Diagnosis and Clarithromycin Resistance Test

For each patient, two gastric biopsies were obtained from the antrum (one sample) and body (one sample). DNA was extracted from biopsy specimens using the QIAamp^®^ DNA Mini Kit (QIAGEN, Hilden, Germany). After DNA extraction and purification, RT-PCR was carried out using a thermal cycler (CFX96; Bio-Rad^®^, Hercules, CA, USA) and employing the Allplex™ *H. pylori & ClaR* Assay kit (Seegene Inc., Seoul, Republic of Korea). This RT-PCR kit is designed for the simultaneous detection of *H. pylori* DNA and the identification of two-point mutations (A2142G and A2143G) in the bacterial 23S ribosomal RNA gene [11]. Amplification was carried out for up to 50 cycles beyond the threshold cycle, and fluorescence data were automatically interpreted using Seegene Viewer software (version 3.0) according to the instructions of the manufacturer (Appendix A).

### 2.3. Helicobacter pylori Eradication Therapy

Based on the RT-PCR results, patients without point mutations were considered infected with *H. pylori* strains susceptible to clarithromycin and were excluded from this study. The detection of the A2142G or A2143G point mutations was defined as the molecular genetic identification of a clarithromycin-resistant *H. pylori* infection, and these patients were invited to participate in this study. All enrolled patients received TA dual therapy consisting of 50 mg of TPZ twice daily and high-dose amoxicillin (1000 mg three times daily or 750 mg four times daily) for 2 weeks. Six weeks after completing eradication therapy, patients were followed up to assess compliance and adverse drug events. The *H. pylori* eradication outcome was confirmed using the urea breath test (UBT) with a cutoff value of 2.5‰, as recommended by the manufacturer. If the UBT result was positive, bismuth quadruple therapy was recommended as a rescue treatment.

### 2.4. Study Outcomes

The primary outcome of this study was *H. pylori* eradication efficacy. In the intention-to-treat (ITT) analysis, patients who were noncompliant or lost to follow-up were considered to have failed the *H. pylori* treatment. A per-protocol (PP) analysis was conducted after excluding patients who did not complete the eradication regimen or were not followed up. The secondary endpoints included patient compliance and adverse drug events, including bitter taste, abdominal pain, nausea or vomiting, diarrhea, and bloating. One physician (J.-H.C.) evaluated treatment compliance and adverse drug events when the patients returned to the outpatient clinic for assessment of eradication success. Compliance with the eradication regimen was defined as >90% medication consumption.

### 2.5. Serum Pepsinogen Measurement

Before *H. pylori* eradication, fasting blood samples were collected from all patients in the morning. Serum was separated by centrifugation at 3000 rpm for 10 min and stored at −80 °C until analysis. Serum pepsinogen (PG) I and PG II concentrations were measured using a latex turbidimetric immunoassay (HiSens; HBI, Anyang, Republic of Korea) according to the manufacturer’s instructions [12]. The PG I/II ratio was subsequently calculated. After completing TA dual therapy, patients revisited the outpatient clinic for follow-up evaluation. At that time, serum PG levels after eradication were remeasured along with the UBT.

### 2.6. Statistical Analysis

To efficiently identify and evaluate potentially effective therapies for *H. pylori*, the protocol of this pilot study allowed the enrollment of up to 50 patients who had completed the treatment regimen [13]. The collected data were analyzed using SPSS software (version 29.0; IBM Corp., Armonk, NY, USA). Continuous variables are presented as mean ± standard deviation and were compared using Student’s *t*-test. Categorical variables are presented as numbers with percentages and were compared using Pearson’s Chi-square test and the linear-by-linear association test. Binary regression analysis was carried out to identify the factors associated with *H. pylori* eradication outcome. A *p*-value < 0.05 was deemed to be statistically significant.

## 3. Results

### 3.1. Study Population

During the study period, 195 patients were diagnosed with *H. pylori* infection by the RT-PCR assay (Figure 1). Based on these results, 122 patients without A2142G and A2143G point mutations were not eligible for this study. The remaining 73 patients infected with clarithromycin-resistant *H. pylori* strains were invited to participate in the study. Three patients with chronic kidney disease, 11 who did not consent to dual therapy, and two who refused *H. pylori* eradication treatment were excluded from the study. Therefore, a total of 57 patients were included in this study. Of these, seven patients were lost to follow-up that was designed to evaluate the outcome of *H. pylori* eradication treatment.

Among the 50 patients who underwent follow-up, 29 achieved *H. pylori* eradication with TA dual therapy while 21 failed to achieve it (Table 1). There were no significant differences between the groups that achieved or failed to achieve *H. pylori* eradication in terms of age, sex, alcohol consumption, smoking status, family history of gastric cancer, body mass index (BMI), body surface area (BSA), cause of the need for eradication, presence of comorbidities, or degree of gastric atrophy (except for history of eradication). The proportion of patients with a history of eradication was significantly lower in the eradication success group compared with the eradication failure group (17.2% vs. 52.4%, respectively; *p* = 0.009).

### 3.2. Treatment Outcomes

In this study, TA dual therapy was administered to 40 patients in the treatment-naïve and 17 patients in the treatment-experienced group. Six patients in the treatment-naïve group and one patient in the treatment-experienced group were lost to follow-up. The total eradication rates in the ITT and PP analyses were 50.9% (*n* = 29/57) and 58.0% (*n* = 29/50), respectively (Table 2). There was a significant difference in *H. pylori* eradication rates between the treatment-naïve and treatment-experienced groups in the ITT and PP analyses (60.0% vs. 29.4%, *p* = 0.035 and 70.6% vs. 31.3%, *p* = 0.009, respectively). All patients who underwent follow-up completed *H. pylori* treatment. The adverse events associated with *H. pylori* eradication therapy included abdominal pain (*n* = 1), diarrhea (*n* = 1), and skin rash (*n* = 2) in four patients (8.0%). All adverse events were mild, and none of the patients discontinued eradication therapy because of adverse drug events.

Table 3 shows a comparison of serum PG I and PG II levels and the PG I/II ratio after *H. pylori* eradication treatment completion between the eradication success and failure groups. Before *H. pylori* eradication treatment, the mean values of serum PG I, PG II, and PG I/II ratios in the eradication success group were 67.1 (±42.1) ng/mL, 27.5 (±16.6) ng/mL, and 2.64 (±1.4), respectively. The mean values of serum PG I, PG II, and PG I/II ratios in the eradication failure group were 70.3 (±26.7) ng/mL, 24.4 (±7.5) ng/mL, and 2.96 (±1.1), respectively. After the completion of TA dual therapy, PG II levels significantly decreased to 10.9 (±5.7) ng/mL and the PG I/II ratio increased to 4.37 (±1.5) in *H. pylori*-eradicated patients. In contrast, there were no significant changes in serum PG II levels or PG I/II ratios in patients in whom TA dual therapy failed.

### 3.3. Factors Affecting Eradication Rate

As shown in Table 4, 16 patients had previously failed *H. pylori* eradication treatment once (*n* = 12), twice (*n* = 3), or four times (*n* = 1). Most first-line treatment regimens used previously were standard triple therapies containing amoxicillin and clarithromycin for 7–14 days. In three patients in whom *H. pylori* eradication had failed twice, second-line treatment regimens consisted of 14-day standard triple therapy (*n* = 2) and 7-day bismuth quadruple therapy (*n* = 1). In one patient who failed *H. pylori* eradication four times, previous *H. pylori* treatment regimens consisted of 7- or 14-day standard triple therapy, 5-day concomitant therapy, or 7-day triple therapy, including tetracycline and levofloxacin.

The proportion of patients who achieved successful *H. pylori* eradication with TA dual therapy are shown in Table 5, and have been categorized according to sex, age, BMI, BSA, drinking status, smoking habits, causes of eradication, presence of comorbidities, gastric atrophy, amoxicillin administration, and history of eradication. Multivariate analysis revealed that treatment-naïve patients were significantly associated with successful *H. pylori* eradication via TA dual therapy (Odds Ratio, 8.679; 95% Confidence Interval [CI], 1.809–41.648; *p* = 0.007).

## 4. Discussion

Amoxicillin is a time-dependent antibiotic, and its effectiveness is closely related to the duration for which the drug concentration remains above the minimum inhibitory concentration (MIC) for *H. pylori* [14]. Thus, administering amoxicillin at least three times daily is necessary to maintain its therapeutic efficacy in TA dual therapy against *H. pylori*. In an early attempt at dual therapy in 1989, Unge et al. combined omeprazole with amoxicillin (750 mg twice daily for 14 days), but the eradication rate was relatively low (62.5%) [15]. This suboptimal outcome was largely attributed to insufficient acid suppression with standard-dose omeprazole and inadequate amoxicillin dosing frequency.

The administration of high-dose second-generation PPIs maintains a sustained high gastric pH, which optimizes the activity of amoxicillin by creating a favorable intragastric environment. When gastric pH rises above 6, *H. pylori* transitions into a replicative phase, during which amoxicillin exerts its bactericidal effect by inhibiting bacterial cell wall synthesis [16]. In 2015, Yang et al. demonstrated that a dual therapy consisting of 20 mg of rabeprazole and 750 mg of amoxicillin four times daily for 14 days achieved a superior *H. pylori* eradication rate of 95.3% in treatment-naïve patients, compared with 85.3% for sequential therapy and 80.7% for clarithromycin-containing triple therapy [17]. In a meta-analysis, high-dose PPI–amoxicillin dual therapy demonstrated an efficacy and patient compliance that was comparable to current guideline-recommended therapies (83.2–87.5% vs. 85.3–90.1% and 94.3% vs. 93.5%, respectively), while generally resulting in fewer adverse effects (12.9% vs. 28.0%, respectively; *p* = 0.0005) [18]. P-CABs, such as vonoprazan, are not affected by the CYP2C19 genotype because they are primarily metabolized by CYP3A4 [19]. A recent meta-analysis of 15 studies involving 4568 patients demonstrated that the pooled eradication rates of vonoprazan–amoxicillin dual therapy were 85.0% in the ITT analysis and 90.0% in the PP analysis [20]. These results support the use of optimized dual therapy in regions with high clarithromycin resistance, particularly when combined with frequent amoxicillin dosing and adequate acid suppression.

Currently, most of the clinical evidence for high-dose amoxicillin dual therapy with PPI or vonoprazan comes from studies conducted in China [21]. Conversely, amoxicillin dual therapy has not consistently achieved acceptable eradication rates outside of China. According to the European Registry, the eradication rate of dual therapy with PPI and amoxicillin (3 g daily) is low, at 51–52% [22]. Similarly, PPI-amoxicillin dual therapy did not provide adequate eradication rates for *H. pylori* in an Irish population [23]. The overall *H. pylori* eradication rate was 56.1%. In the U.S. and Europe, Chey et al. reported that vonoprazan-amoxicillin dual therapy achieved an eradication rate of 77.2% in all patients and 69.6% in those infected with clarithromycin-resistant *H. pylori* strains [24]. In South Korea, the efficacy of a 2-week high-dose PPI-amoxicillin dual therapy has been evaluated in only a single study, which reported an overall eradication rate of 52% [25]. Thus, current evidence related to PPI–amoxicillin dual therapy does not meet the satisfactory eradication threshold recommended by the South Korean guidelines, which is >85–90% for empirical first-line treatment [26].

TPZ, launched in South Korea in 2018, has shown noninferiority to conventional PPIs in the treatment of reflux esophagitis, peptic ulcer disease, and post-endoscopic resection ulcers [27]. In China, Liu et al. conducted the first randomized controlled trial investigating TA dual therapy for *H. pylori* infection and compared it with a modified bismuth quadruple therapy comprising amoxicillin and clarithromycin [28]. This study demonstrated comparable *H. pylori* eradication rates in the two treatments (83.9–88.3% vs. 81.4–84.8%, respectively). In two additional studies, TA dual therapy demonstrated similar *H. pylori* eradication rates, with 85.8–86.0% in the ITT analysis and 88.2–93.8% in the PP analysis [29,30]. The incidence of adverse drug events was significantly lower in the dual therapy group than in the modified bismuth quadruple therapy group (14.6% vs. 27.2%, respectively; *p* = 0.026). However, no data are available regarding the efficacy and safety of TA dual therapy in the South Korean population.

Molecular testing has been increasingly established and widely utilized in South Korea to detect bacterial 23S rRNA point mutations associated with clarithromycin-resistant *H. pylori* strains [31]. For *H. pylori* strains confirmed as clarithromycin-susceptible by molecular testing, a 1-week standard triple therapy achieved eradication rates > 90%, similar to the 14-day regimen [32]. Although the bismuth quadruple regimen is effective against clarithromycin-resistant *H. pylori* strains, its clinical utility is often limited by frequent adverse events, such as nausea, vomiting, abdominal pain, and diarrhea [33]. In contrast, amoxicillin dual therapy was well tolerated by most patients, with a low incidence of adverse events [34]. From an antimicrobial stewardship perspective in the management of infectious diseases, dual therapy with a single antibiotic guided by molecular testing may represent an ideal strategy for *H. pylori* treatment [35].

This is the first study to evaluate the efficacy of TA dual therapy as an alternative treatment for clarithromycin-resistant *H. pylori*. However, the results were unsatisfactory, with overall eradication rates ranging from 50.9 to 58.0%, which were lower than expected. In treatment-experienced patients, the present study showed unacceptably low eradication rates of 29.4–31.3%, consistent with an Irish study, which reported an eradication rate of 43.5% [23]. This may be attributed to the development of secondary amoxicillin resistance because all patients in this study had previously received standard triple therapy containing amoxicillin. Nishizawa et al. reported that the rate of amoxicillin resistance increased after unsuccessful *H. pylori* eradication therapy [36]. The MIC_90_ of amoxicillin doubled with each subsequent eradication failure. The rate of resistance to amoxicillin in the group with a history of three prior eradication failures was 18.2%.

There are two possible explanations for the suboptimal eradication rates observed in this study. First, the actual rate of resistance to amoxicillin in *H. pylori* treatment-naïve patients may be higher than that reported previously. Antibiotic resistance rates vary depending on the region, population, and MIC breakpoints used (e.g., European Committee on Antimicrobial Susceptibility Testing [EUCAST] vs. local standards). In South Korea, previous studies have often used a higher MIC breakpoint for amoxicillin resistance in *H. pylori*, typically defined as MIC ≥ 0.5 µg/mL, rather than the more stringent EUCAST standard of MIC > 0.125 µg/mL [37]. When the EUCAST breakpoint was retrospectively applied to previous data, the reported amoxicillin resistance rates increased from 9.0–9.7% to 21.1–21.4% [38,39]. In a study by Chen et al., amoxicillin resistance (MIC ≥ 0.5 µg/mL) was significantly associated with treatment failure of amoxicillin-containing regimens (pooled risk ratio, 1.41; 95% CI, 1.12–1.78; *p* = 0.004) [40]. Therefore, standardizing the MIC breakpoints for defining amoxicillin resistance is essential for optimizing treatment outcomes, particularly for dual therapies that rely solely on amoxicillin.

Second, the 50 mg twice daily TPZ dose may be insufficient for effective *H. pylori* eradication. TPZ-based regimens using 50 mg twice daily did not demonstrate superior *H. pylori* eradication rates compared with PPI-based treatments (ITT, 77.3% vs. 76.4%, *p* = 0.68; PP, 84.3% vs. 84.2%, *p* = 0.69) [41]. In a recent study involving treatment-naïve patients with *H. pylori* infections receiving 10-day standard triple therapy, the dosage of TPZ was increased from 50 mg to 100 mg twice daily [42]. Notably, the eradication rate in the 100 mg TPZ group was 86.7%, which was significantly higher than the 66.7% observed in the 50 mg group and comparable to the 87.5% rate achieved with a vonoprazan 20 mg twice daily regimen. Therefore, in future, a TPZ dose escalation may be necessary to improve *H. pylori* treatment outcomes.

In this study, amoxicillin was administered as either 1000 mg three times daily or 750 mg four times daily, according to patient preference. Before prescribing, all patients were instructed that the regimen should maintain continuous exposure to amoxicillin. Consequently, the eradication efficacy of TA dual therapy did not differ between the two dosing schedules, both of which provided a total daily dose of 3 g. These findings were consistent with those reported by Qiu et al., demonstrating that 1000 mg three times daily was non-inferior to 750 mg four times daily in vonoprazan–amoxicillin dual therapy (ITT, 89.9% vs. 93.6%; PP, 94.1% vs. 99.0%) [43]. Considering patient convenience and comparable efficacy, TA dual therapy with 1000 mg of amoxicillin administered three times daily appears to be the more practical regimen.

The present study demonstrated changes in serum PG levels, specifically a decrease in PG II and an increase in the PG I/II ratio, following successful *H. pylori* eradication with TA dual therapy. In a previous study, successful eradication using standard triple therapy was also significantly associated with decreased serum PG II (from 25.4 to 9.1 ng/mL, *p* < 0.001) and increased PG I/II ratio (from 3.07 to 4.98, *p* < 0.001) [44]. These findings suggest that recovery of serum PG levels can also be achieved through TA dual therapy, indicating effective restoration of gastric mucosal function after eradication.

This study has several limitations. First, it was a single-arm pilot study without a comparator group. The results cannot be generalized until large-scale multicenter comparative trials have been conducted to confirm the efficacy of TA dual therapy as an alternative to guideline-recommended regimens. Second, among the antibiotic resistance tests, only molecular testing for clarithromycin resistance was carried out. The eradication success rate for *H. pylori* in relation to amoxicillin resistance was not evaluated, which limits the interpretation of treatment efficacy in resistant strains. Finally, intragastric pH levels and plasma amoxicillin concentrations were not measured, preventing an assessment of the achievement of sufficient acid suppression and antibiotic exposure during dual therapy.

## 5. Conclusions

The eradication efficacy of TA dual therapy against clarithromycin-resistant *H. pylori* strains was suboptimal. However, our treatment regimen demonstrated favorable safety outcomes, with a low incidence of adverse events and high treatment compliance. Notably, eradication rates were significantly lower in patients with prior *H. pylori* treatment than in treatment-naïve individuals. Further investigations are warranted to determine whether this dual therapy can serve as an alternative regimen for patients with confirmed clarithromycin-resistant *H. pylori* infections.

## Figures and Tables

**Figure 1 microorganisms-13-02408-f001:**
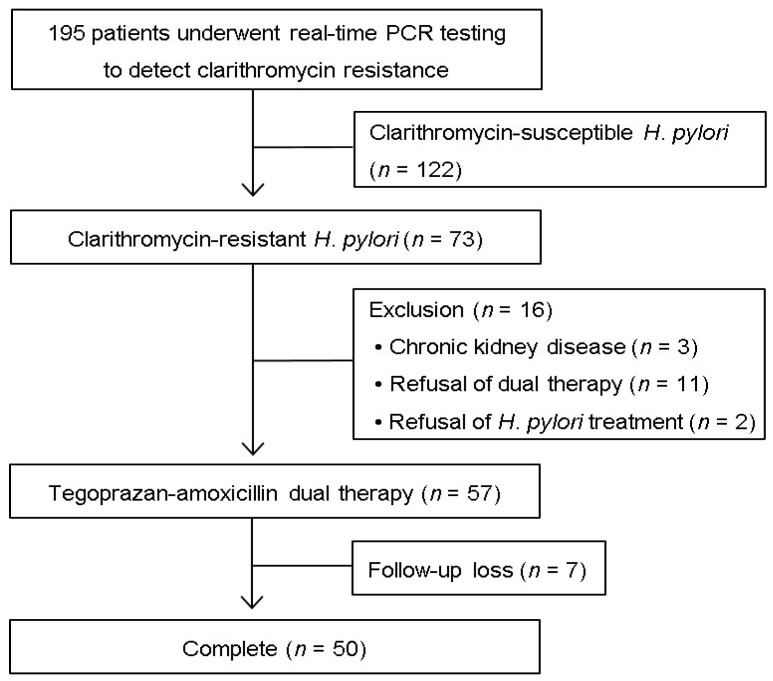
Flow chart showing patient enrollment.

**Table 1 microorganisms-13-02408-t001:** Baseline characteristics of the study population.

	Eradication Success (*n* = 29)	Eradication Failure (*n* = 21)	Total (*n* = 50)	*p*-Value *
Age, years, mean (SD)	63.2 (18.1)	62.9 (12.3)	63.1 (15.8)	0.942
Male (%)	15 (51.7)	11 (52.4)	26 (52.0)	0.963
Alcohol drinker (%)	16 (55.2)	10 (47.6)	26 (52.0)	0.598
Smoking status (%)	10 (34.5)	10 (47.6)	20 (40.0)	0.349
Family history of gastric cancer (%)	3 (10.3)	4 (19.0)	7 (14.0)	0.434
Body mass index, kg/m^2^, mean (SD)	24.3 (3.2)	24.1 (2.2)	24.2 (2.8)	0.787
Body surface area, m^2^, mean (SD)	1.66 (0.2)	1.71 (0.2)	1.68 (0.2)	0.380
Cause of need for eradication (%)				0.201
Chronic active gastritis	14 (48.3)	14 (66.7)	28 (56.0)	
Peptic ulcer/neoplasia	15 (51.7)	7 (33.3)	22 (44.0)	
Comorbidity (%)	14 (48.3)	12 (57.1)	26 (52.0)	0.536
Degree of gastric atrophy ** (%)				0.490
Closed-type	18 (62.1)	15 (71.4)	33 (66.0)	
Open-type	11 (37.9)	6 (28.6)	17 (34.0)	
History of eradication (%)	5 (17.2)	11 (52.4)	16 (32.0)	0.009

SD, standard deviation. * Successful eradication vs. failure. ** Atrophy is sub-grouped by the Kimura-Takemoto classification.

**Table 2 microorganisms-13-02408-t002:** Eradication rate, compliance, and adverse events of Tegoprazan–Amoxicillin dual therapy.

	Treatment-Naïve	Treatment-Experienced	Total	*p*-Value *
Intention-to-treat analysis				
Eradication rate	60.0% (24/40)	29.4% (5/17)	50.9% (29/57)	0.035
Per-protocol analysis				
Eradication rate	70.6% (24/34)	31.3% (5/16)	58.0% (29/50)	0.009
Compliance	100% (34/34)	100% (16/16)	100% (50/50)	N/A
Adverse events	5.9% (2/34)	12.5% (2/16)	8.0% (4/50)	0.584

* Treatment-naïve vs. treatment-experienced.

**Table 3 microorganisms-13-02408-t003:** Changes in serum pepsinogen levels after Tegoprazan–Amoxicillin dual therapy.

	Eradication Success (*n* = 29)			Eradication Failure (*n* = 21)		
	Before	After	*p*-Value	Before	After	*p*-Value
Serum PG (mean ± SD)						
PG I, ng/mL	67.1 ± 42.1	50.5 ± 51.2	0.099	70.3 ± 26.7	70.6 ± 29.1	0.956
PG II, ng/mL	27.5 ± 16.6	10.9 ± 5.7	<0.001	24.4 ± 7.5	24.8 ± 11.1	0.872
PG I/II ratio	2.64 ± 1.4	4.37 ± 1.5	<0.001	2.96 ± 1.1	3.17 ± 1.8	0.521

PG, pepsinogen; SD, standard deviation.

**Table 4 microorganisms-13-02408-t004:** Clinical data of patients receiving Tegoprazan–Amoxicillin dual therapy as rescue treatment.

Times of Previous Eradication	1	2	4	Total
Number of patients	12	3	1	16
BMI, kg/m^2^, mean (SD)	24.3 (2.4)	22.8 (2.6)	25.5	24.1 (2.4)
BSA, m^2^, mean (SD)	1.60 (0.2)	1.79 (0.2)	1.72	1.64 (0.2)
Alcohol drinker (%)	6 (50.0)	1 (33.3)	0	7 (43.8)
Smoking status (%)	5 (41.7)	1 (33.3)	0	6 (37.5)
Antibiotics used in previous eradication treatment	AMX, CAM	AMX, CAM, MDZ, TET	AMX, CAM, MDZ, TET, LVFX	AMX, CAM, MDZ, TET, LVFX

BMI, body mass index; BSA; body surface area; SD; standard deviation, AMX, amoxicillin; CAM, clarithromycin; MDZ, metronidazole; TET, tetracycline; LVFX, levofloxacin.

**Table 5 microorganisms-13-02408-t005:** Factors associated with successful eradication of Tegoprazan–Amoxicillin dual therapy.

	Eradication Success	Multivariate Analyses		
		OR	95% CI	*p* Value
Sex				
Female (*n* = 24)	14 (58.3)	1 (ref.)		
Male (*n* = 26)	15 (57.7)	3.029	0.220–41.631	0.407
Age, years				
<65 (*n* = 23)	14 (60.9)	1 (ref.)		
≥65 (*n* = 27)	15 (55.6)	0.360	0.049–2.637	0.315
Body mass index, kg/m^2^				
<22.4 (*n* = 14)	8 (57.1)	1 (ref.)		
≥22.4 (*n* = 36)	21 (58.3)	0.850	0.106–6.834	0.878
Body surface area, m^2^				
<1.72 (*n* = 32)	20 (62.5)	1 (ref.)		
≥1.72 (*n* = 18)	9 (50.0)	0.227	0.027–1.939	0.176
Alcohol drinking				
No (*n* = 24)	13 (54.2)	1 (ref.)		
Yes (*n* = 26)	16 (61.5)	2.198	0.249–19.440	0.479
Current smoker				
No (*n* = 30)	19 (63.3)	1 (ref.)		
Yes (*n* = 20)	10 (50.0)	0.079	0.005–1.281	0.074
Causes of *H. pylori* eradication				
Chronic active gastritis (*n* = 28)	14 (50.0)	1 (ref.)		
Peptic ulcer/neoplasia (*n* = 22)	15 (68.2)	6.299	0.988–40.167	0.052
Comorbidity				
Absent (*n* = 24)	15 (62.5)	1 (ref.)		
Present (*n* = 26)	14 (53.8)	0.637	0.108–3.759	0.618
Gastric atrophy				
Closed-type (*n* = 33)	18 (54.5)	1 (ref.)		
Open-type (*n* = 14)	11 (64.7)	2.899	0.505–16.649	0.233
Amoxicillin administration				
1 g TID (*n* = 27)	15 (55.6)	1 (ref.)		
750 mg QID (*n* = 23)	14 (60.9)	1.133	0.242–5.307	0.874
History of eradication				
Treatment-experienced (*n* = 16)	5 (31.3)	1 (ref.)		
Treatment-naïve (*n* = 34)	24 (70.6)	8.679	1.809–41.648	0.007

*H. pylori*, *Helicobacter pylori*; OR, odds ratio; CI, confidence interval.

## Data Availability

The data presented in this study are available on request from the author due to patients data is difficult to make publicly available.

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
