# Peer review of "Tegoprazan–Amoxicillin Dual Therapy for Clarithromycin-Resistant *Helicobacter pylori*: A Feasibility Pilot Study"

_microorganisms, 2025, doi:10.3390/microorganisms13102408_

Round 1

Reviewer 1 Report

Comments and Suggestions for Authors

I read the presented pilot study with great interest. Initially, I was surprised that the author was examining the effectiveness of tegoprazan and high-dose amoxicillin therapy, taking into account resistance to clarithromycin, which is not used in the treatment regimen being studied. However, the author convinced me of the feasibility of this study, as it reflects a possible additional marker of treatment failure in the context of high clarithromycin resistance worldwide and the potential for a transition to dual therapy in many regions. Of course, it would be important to understand the impact of amoxicillin resistance on eradication effectiveness, but the author himself notes this limitation of the study. Indeed, this data would be valuable to obtain. Since the author uses two dosing options when prescribing 3 grams of amoxicillin—750 mg 4 times a day and 1000 mg 3 times a day—I lacked an explanation for why one of these dosing approaches was not used, nor how this affected compliance, adverse events, and treatment efficacy. Only Table 5 provides a comparison of these two approaches. I would request a more complete and detailed analysis. This is an important aspect of the study, but it is insufficiently addressed.
Starting with line 159 and in Table 3, the author analyzes pepsinogen I and II levels and their ratio, yet the materials and methods section does not mention the use of these methods. The data on the increase of pepsinogen I/II racio after successful eradication indicate an improvement in gastric function. However, its relevance to the study objective is unclear.

Author Response

Reviewer: 1

  1. I read the presented pilot study with great interest. Initially, I was surprised that the author was examining the effectiveness of tegoprazan and high-dose amoxicillin therapy, taking into account resistance to clarithromycin, which is not used in the treatment regimen being studied. However, the author convinced me of the feasibility of this study, as it reflects a possible additional marker of treatment failure in the context of high clarithromycin resistance worldwide and the potential for a transition to dual therapy in many regions. Of course, it would be important to understand the impact of amoxicillin resistance on eradication effectiveness, but the author himself notes this limitation of the study. Indeed, this data would be valuable to obtain. Since the author uses two dosing options when prescribing 3 grams of amoxicillin—750 mg 4 times a day and 1000 mg 3 times a day—I lacked an explanation for why one of these dosing approaches was not used, nor how this affected compliance, adverse events, and treatment efficacy. Only Table 5 provides a comparison of these two approaches. I would request a more complete and detailed analysis. This is an important aspect of the study, but it is insufficiently addressed.

Answer: Thank you for your valuable comment. Regardless of the dosing schedule—whether 1000 mg three times daily or 750 mg four times daily—no significant differences were observed in eradication efficacy, patient compliance, or adverse drug events. Based on your comment, the following paragraph was added to the Discussion section as shown below.

In this study, amoxicillin was administered as either 1000 mg three times daily or 750 mg four times daily, according to patient preference. Before prescribing, all patients were instructed that the regimen should maintain continuous exposure to amoxicillin. Consequently, the eradication efficacy of TA dual therapy did not differ between the two dosing schedules, both of which provided a total daily dose of 3 g. These findings were consistent with those reported by Qiu et al., demonstrating that 1000 mg three times daily was non-inferior to 750 mg four times daily in vonoprazan-amoxicillin dual therapy (ITT, 89.9% vs. 93.6%; PP, 94.1% vs. 99.0%) [43]. Considering patient convenience and comparable efficacy, TA dual therapy with 1000 mg of amoxicillin administered three times daily appears to be the more practical regimen.

  1. Starting with line 159 and in Table 3, the author analyzes pepsinogen I and II levels and their ratio, yet the materials and methods section does not mention the use of these methods. The data on the increase of pepsinogen I/II racio after successful eradication indicate an improvement in gastric function. However, its relevance to the study objective is unclear.

Answer: Thank you for your careful comment. In response to your comment, the serum pepsinogen measurement has been added to the Methods section.

2.5. Serum Pepsinogen Measurement

Before H. pylori eradication, fasting blood samples were collected from all patients in the morning. Serum was separated by centrifugation at 3,000 rpm for 10 minutes and stored at −80°C until analysis. Serum pepsinogen (PG) I and PG II concentrations were measured using a latex turbidimetric immunoassay (HiSens; HBI, Anyang, South Korea) according to the manufacturer’s instructions [12]. The PG I/II ratio was subsequently calculated. After completing TA dual therapy, patients revisited the outpatient clinic for follow-up evaluation. At that time, serum PG levels after eradication were remeasured along with the UBT.

Its clinical implications have been addressed in the Discussion section as suggested.

The present study demonstrated changes in serum PG levels, specifically a decrease in PG II and an increase in the PG I/II ratio, following successful H. pylori eradication with TA dual therapy. In a previous study, successful eradication using standard triple therapy was also significantly associated with decreased serum PG II (from 25.4 to 9.1 ng/mL, p < 0.001) and increased PG I/II ratio (from 3.07 to 4.98, p < 0.001) [44]. These findings suggest that recovery of serum PG levels can also be achieved through TA dual therapy, indicating effective restoration of gastric mucosal function after eradication.

Reviewer 2 Report

Comments and Suggestions for Authors

This study aimed to investigate the efficacy and safety of TPZ-amoxicillin (TA) dual therapy against clarithromycin-resistant H. pylori strains. However, the results were unsatisfactory, with overall eradication rates ranging from 50.9% to 58.0%. A few concerns were addressed by the authors in the limitations of the study.

- Only 57 patients were included in this study, but 50 completed the study.

- A bit of clarification on the following. The authors described that the TA dual therapy was administered to 40 patients in the treatment-naïve group and 17 patients in the treatment-experienced group. Do you mean no previous amoxicillin exposure (treatment-naïve) and previous exposure and failure in the treatment-experienced group?

- The authors did not expand on the results from the pepsinogen levels, which clearly indicated eradication success. There were no significant changes in serum PG II levels or PG I/II ratios in patients in whom TA dual therapy failed.

-  I liked the idea that in areas where known rates of resistance to some antibiotics are known, an antibiotic guided by molecular testing may represent an ideal strategy for H. pylori treatment.

Author Response

Reviewer: 2

  1. Only 57 patients were included in this study, but 50 completed the study.

Answer: Thank you for your careful comment. This study was designed as a pilot trial to evaluate the preliminary efficacy and safety of tegoprazan–amoxicillin (TA) dual therapy in South Korea. As noted in reference #12 by Graham D.Y., a pilot study enables the assessment of therapeutic potential under maximally effective conditions while ensuring patient safety. The findings from this trial are expected to inform subsequent decisions regarding optimal dosing regimens and the feasibility of conducting larger confirmatory clinical trials. Accordingly, as described in the statistical analysis section, the sample size was intentionally limited to 50 patients who had completed the treatment regimen and subsequently visited the outpatient clinic for follow-up evaluation. The remaining seven patients did not return to the outpatient clinic for assessment of the eradication outcome, and thus were excluded from the per-protocol analysis.

To efficiently identify and evaluate potentially effective therapies for H. pylori, the protocol of this pilot study allowed the enrollment of up to 50 patients who had completed the treatment regimen [13].

  1. A bit of clarification on the following. The authors described that the TA dual therapy was administered to 40 patients in the treatment-naïve group and 17 patients in the treatment-experienced group. Do you mean no previous amoxicillin exposure (treatment-naïve) and previous exposure and failure in the treatment-experienced group?

Answer: Thank you for your valuable comment. Yes, your understanding is correct. The treatment-naïve group was defined as patients who had never received H. pylori eradication therapy, whereas the treatment-experienced group included those who had undergone such treatment at least once in the past. As shown in Table 4, amoxicillin was included among the antibiotics used in all previous eradication treatments.

  1. The authors did not expand on the results from the pepsinogen levels, which clearly indicated eradication success. There were no significant changes in serum PG II levels or PG I/II ratios in patients in whom TA dual therapy failed.

Answer: Thank you for your kind comment. In the revised manuscript, the clinical implications of the recovery of serum PG II and the PG I/II ratio have been addressed in the Discussion section, as shown below.

The present study demonstrated changes in serum PG levels, specifically a decrease in PG II and an increase in the PG I/II ratio, following successful H. pylori eradication with TA dual therapy. In a previous study, successful eradication using standard triple therapy was also significantly associated with decreased serum PG II (from 25.4 to 9.1 ng/mL, p < 0.001) and increased PG I/II ratio (from 3.07 to 4.98, p < 0.001) [44]. These findings suggest that recovery of serum PG levels can also be achieved through TA dual therapy, indicating effective restoration of gastric mucosal function after eradication.

  1. I liked the idea that in areas where known rates of resistance to some antibiotics are known, an antibiotic guided by molecular testing may represent an ideal strategy for H. pylori treatment.

Answer: Thank you for your careful comment. In this study, TA dual therapy demonstrated excellent treatment compliance and a favorable safety profile in patients infected with clarithromycin-resistant H. pylori strains, as determined by real-time PCR testing. Although the eradication rate was suboptimal, TA dual therapy may still represent a viable treatment option for elderly patients with comorbidities, given its good tolerability and simplicity of administration.  

Round 2

Reviewer 1 Report

Comments and Suggestions for Authors

The author has taken into account the comments and supplemented the article in accordance with the recommendations. It may be published.